# Serum Neutralizing Activity against B.1.1.7, B.1.351, and P.1 SARS-CoV-2 Variants of Concern in Hospitalized COVID-19 Patients

**DOI:** 10.3390/v13071347

**Published:** 2021-07-12

**Authors:** Claudia Maria Trombetta, Serena Marchi, Simonetta Viviani, Alessandro Manenti, Linda Benincasa, Antonella Ruello, Emilio Bombardieri, Ilaria Vicenti, Maurizio Zazzi, Emanuele Montomoli

**Affiliations:** 1Department of Molecular and Developmental Medicine, University of Siena, 53100 Siena, Italy; serena.marchi2@unisi.it (S.M.); simonetta.viviani@unisi.it (S.V.); emanuele.montomoli@unisi.it (E.M.); 2VisMederi srl, Strada del Petriccio e Belriguardo 35, 53100 Siena, Italy; alessandro.manenti@vismederi.com; 3VisMederi Research srl, Strada del Petriccio e Belriguardo 35, 53100 Siena, Italy; linda.benincasa@vismederiresearch.com; 4Humanitas Gavazzeni, Via Mauro Gavazzeni 21, 24125 Bergamo, Italy; antonella.ruello@gavazzeni.it (A.R.); emilio.bombardieri@gavazzeni.it (E.B.); 5Department of Medical Biotechnologies, University of Siena, 53100 Siena, Italy; ilariavicenti@gmail.com (I.V.); maurizio.zazzi@gmail.com (M.Z.)

**Keywords:** neutralizing activity, SARS-CoV-2, variants of concern

## Abstract

The recent spreading of new SARS-CoV-2 variants, carrying several mutations in the spike protein, could impact immune protection elicited by natural infection or conferred by vaccination. In this study, we evaluated the neutralizing activity against the viral variants that emerged in the United Kingdom (B.1.1.7), Brazil (P.1), and South Africa (B.1.351) in human serum samples from hospitalized patients infected by SARS-CoV-2 during the first pandemic wave in Italy in 2020. Of the patients studied, 59.5% showed a decrease (≥2 fold) in neutralizing antibody titer against B.1.1.7, 83.3% against P.1, and 90.5% against B.1.351 with respect to the original strain. The reduction in antibody titers against all analyzed variants, and in particular P.1 and B.1.351, suggests that previous symptomatic infection might be not fully protective against exposure to SARS-CoV-2 variants carrying a set of relevant spike mutations.

## 1. Introduction

One year ago, the Director-General of the World Health Organization declared the first pandemic caused by a coronavirus named severe acute respiratory syndrome coronavirus 2 (SARS-CoV-2) [1]. Neutralizing antibodies, targeting the viral spike (S) protein and its receptor-binding domain (RBD), are considered a surrogate of protection against COVID-19 [2], the disease caused by SARS-CoV-2, although no formal correlates of protection have been established so far. In fact, available COVID-19 vaccines that have shown high efficacy are developed with the concept that the S protein is the immunodominant antigen. These vaccines are designed based on the Wuhan strain, the original strain that since January 2020 has been at the origin of the worldwide pandemic.

In the last months, different variants of concern (VOCs) [3] of SARS-CoV-2 emerged around the world, initially as occasional isolates; however, in some settings, the most efficient variants are rapidly replacing the original Wuhan strain.

One new variant emerged in the United Kingdom (UK), affecting people under 60 years of age. Retrospective analyses have dated the first identification as occurring in September 2020 in South East England [4,5]. This variant, called B.1.1.7, spread in several countries around the world (i.e., Italy, Denmark, the United States, the Netherlands, Australia, Iceland), and it has been associated with 50% increased transmission [6] and increased risk of death [4,7,8,9]. Seventeen mutations/deletions in the viral genome characterize the B.1.1.7, including eight in the S protein (Δ69–70, ΔY144, N501Y, A570D, P681H, T716I, S982A, D118H) [5,8]. Three mutations are of biological significance, namely Δ69–70, N501Y, and P681H. The first one is related to an increased viral infectivity and a potential impact on PCR assays targeting the S gene [4,8,10]; however, this mutation is not restricted to this variant [10]. The N501Y is a mutation in the RBD resulting in an increased binding affinity to its cellular ACE-2 receptor [7,8,11]. The last one, P681H, is supposed to be related to an improved transmissibility of the virus [12].

Two new variants emerged in Brazil, both originating from the B.1.1.28 clade [13]. The first variant (named B.1.1.28.2), detected in October 2020, was characterized by the E484K mutation in the S protein, related to a possible escape from neutralizing antibodies [14]. The second variant (named B.1.1.28.1) was reported by the National Institute of Infectious Diseases in Japan after sampling of Brazilian travelers to Japan. Since this variant presents other substantial mutations such as H655Y, L18F, D138Y, and N501Y, in addition to the E484K [7,13,15], it was classified as a “variant of concern” [16]. Notably, the E484K mutation is in common with the B.1.351 variant first identified in the Republic of South Africa but not with B.1.1.7 [13]. The P.1 and B.1.351 variants share common mutations in the S protein, increasing the possibility of evasion of the humoral response and enhanced transmissibility [13,17].

As of 2 March 2021, B.1.1.7 accounts for 54.0% of Italian cases nationwide, ranging from 0 to 93.3% between regions, and is becoming the most widely present variant in the country. The P.1 variant is less prevalent for now, accounting for 4.3% of all new local COVID-19 cases, while B.1.351 is involved in just 0.4% of new cases [18]. Recently, a new VOC emerged, the SARS-CoV-2 B.1.617.2, also named Delta, first detected in India and now dominant in the UK [19,20]. The new variant has overtaken B.1.1.7 in the UK, and it is estimated that it will represent 90% of all viruses circulating in Europe by the end of August [19,21]. Additionally, it seems to be characterized by a higher risk of hospital admission and increased transmissibility [19,20,22]. Here we assess the neutralization activity against B.1.1.7, P.1, and B.1.351 variants in a panel of human serum samples from hospitalized infected COVID-19 patients, previously tested by 2019-nCov/Italy-INMI1, clade V strain (Wuhan strain).

## 2. Materials and Methods

### 2.1. Study Population

Sera of 42 COVID-19 patients (35 recovered and 7 with fatal outcome), hospitalized at Humanitas Gavazzeni (Bergamo, Italy) during the first epidemic wave that occurred in Italy between March and May 2020, were included in the present study. Subject characteristics and study procedures were described in detail elsewhere (approval number 17373—Ethics Committee of the University of Siena; approval number 17/20—Ethics Committee of Humanitas Gavazzeni) [23].

For the purpose of the present study, sera available for each patient at three time points were selected: the hospital admission sample (baseline), the sample showing the highest neutralizing antibody titer against 2019-nCov/Italy-INMI1 strain (hereafter referred to as wild-type (wt) strain) found in the previous study [23] and defined as “peak”, and the last sample available during hospital stay. For 22 patients, the “peak” sample was the last sample available during the hospital stay. Samples of deceased/recovered patients were pulled together for the purpose of this study, as in the original study no difference was found between the 2 groups in terms of neutralizing antibody titers with respect to the wt strain.

### 2.2. Cell Culture and Viral Growth

VERO E6 cells (ATCC—CRL 1586) were cultured in Dulbecco’s Modified Eagle’s Medium (DMEM), High Glucose (Euroclone, Pero, Italy), supplemented with 2 mM L-glutamine (Lonza, Milano, Italy), 100 units/mL penicillin–streptomycin mixture (Lonza, Milano, Italy) and 10% fetal bovine serum (FBS) (Euroclone, Pero, Italy), in a 37 °C and 5% CO_2_ humidified incubator.

Adherent subconfluent cell monolayers of VERO E6 were prepared in growth medium (DMEM High Glucose containing 2% FBS, 2 mM L-glutamine, 100 units/mL penicillin–streptomycin) in 175 cm^2^ flasks or 96-well plates for propagation or titration and neutralization tests of SARS-CoV-2, respectively.

Cells were seeded in a 175 cm^2^ flask at a density of 1 × 10^6^ cells/mL. After 18–20 h, the subconfluent cell monolayer was washed twice with sterile Dulbecco’s phosphate-buffered saline (DPBS). After the DPBS was removed, cells were infected with 3.5 mL of DMEM 2% FBS containing the SARS-CoV-2 virus at a multiplicity of infection (MOI) of 0.01. After 1 h of incubation at 37 °C in a humidified atmosphere with 5% CO_2_, 50 mL of DMEM containing 2% FBS was added. The flasks were observed daily, and the virus was harvested when 80–90% of the cells manifested cytopathic effect (CPE). The culture medium was centrifuged at +4 °C and 469× *g* for 8 min, aliquoted, and stored at −80 °C.

### 2.3. SARS-CoV-2 Viruses

The SARS-CoV-2 (nCoV strain 2019-nCov/Italy-INMI1 strain) wt virus was purchased from the European Virus Archive goes Global (EVAg, Spallanzani Institute, Rome). Notably, the used strain did not carry the S protein amino acid change D614G.

The B.1.1.7 named England/MIG457/2020 and the B.1.351 variant named hCoV-19/Netherlands/NoordHolland_10159/2021, next strain clade 20H, wt viruses were purchased from EVAg.

The P.1 variant (next strain 20J/501Y.V3) (lineage B.1.1.28.1) was kindly provided by the University of Siena, Department of Medical Biotechnology.

### 2.4. Virus Neutralization Assay

The virus neutralization (VN) assay was performed as previously reported [24]. Briefly, after heat-inactivation for 30 min at 56 °C, serum samples, starting from 1:10 dilution, were mixed with an equal volume of SARS-CoV-2 (B.1.1.7, P.1, and B.1.351) viral solution containing 100 tissue culture infective dose 50% (TCID50). After 1 h of incubation at room temperature, 100 µL of virus–serum mixture was added to a 96-well plate containing an 80% confluent Vero E6 cell monolayer. Plates were incubated for 4 days at 37 °C and 5% CO_2_ in humidified atmosphere and then checked for presence/absence of CPE by an inverted optical microscope. A CPE higher than 50% indicated infection. The VN titer was expressed as the reciprocal of the highest serum dilution showing protection from viral infection and CPE.

### 2.5. Statistical Analysis

All statistical analyses were performed using GraphPad Prism version 6.01 for Windows (GraphPad Software, San Diego, CA, USA [25]. For each variant, the mean fold decrease in antibody levels with respect to the *wt* strain was calculated along with its standard deviation (SD). Antibody levels were expressed as log and statistically evaluated with respect to the wt strain using a paired *t*-tests. Statistical significance was set at *p* < 0.05, two-tailed.

## 3. Results

Neutralizing antibody titers of each patient by time point and variant are shown in Figure 1 with statistically significant titer decrease for all three variants at any time point at hospital admission (baseline); 10 patients (23.8%) were negative for neutralizing antibodies against the wt strain as well as for all the variants. In total, 16 patients (38.1%) were negative for B.1.1.7, 20 (47.6%) were negative for P.1, and 30 (71.4%) were negative for B.1.351 at baseline. Twenty-three (54.8%) and 30 (71.4%) patients showed a ≥2-fold decrease in neutralizing antibody titer when tested against B.1.1.7 and P.1, respectively (Table 1a,b). Thirty-two patients (76.2%) showed a ≥2-fold decrease in neutralizing antibody titer against B.1.351 (Table 1c). A significant decrease in neutralizing antibody titers against all three variants (*p* < 0.0001) was observed with respect to the wt strain.

Twenty-six samples (61.9%) with peak neutralizing titers for the wt strain had a ≥2-fold decrease in neutralizing antibody titer against B.1.1.7, with a mean decrease of 3.5-fold (SD 5.0). In particular, 14 (33.3%) showed a 2-fold decrease, 8 (19.0%) showed a 4-fold decrease, and 4 (9.5%) showed a >4-fold decrease (Table 1a). When tested for P.1, 37 (88.1%) had a ≥2-fold decrease with a mean decrease of 6.0-fold (SD 4.3). Ten (23.8%) showed a 2-fold decrease, 12 (28.6%) showed a 4-fold decrease, and 15 (35.7%) showed a >4-fold decrease (Table 1b). When tested for B.1.351, 38 (90.5%) showed a ≥2-fold decrease with a mean decrease of 18.5-fold (SD 18.9); 5 (11.9%) showed a 2-fold decrease, 5 (11.9%) showed a 4-fold decrease, and 28 (66.7%) showed a ≥4-fold decrease (Table 1c). The decrease in neutralizing antibody titer with respect to the wt strain was significant for all three variants (*p* < 0.0001).

At discharge/decease, all patients showed neutralizing antibody against the wt strain, although a decline in titer was observed as deeply described elsewhere [23]. When tested against B.1.1.7, one patient (2.4%) was found negative. A ≥2-fold decrease was observed in 25 (59.5%) patients: 16 (38.1%) and 9 (21.4%) showed a 2-fold and 4-fold decrease, respectively (mean decrease 2.3-fold (SD 1.4)) (Table 1a). Three patients (7.1%) were found negative when tested against P.1, and a ≥2-fold decrease was observed in 35 (83.3%) patients. In particular, 14 (33.3%) showed a 2-fold decrease, 14 (33.3%) showed a 4-fold decrease, and 7 (16.7%) showed a >4-fold decrease, with a mean decrease of 4.4-fold (SD 3.4) (Table 1b).

Nine patients (21.4%) were found negative when tested against B.1.351, and a ≥2-fold decrease was observed in 38 (90.5%) patients. In particular, 9 (21.4%) showed a 2-fold decrease, 6 (14.3%) showed a 4-fold decrease, and 23 (54.8%) showed a >4-fold decrease, with a mean decrease of 11.0-fold (SD 12.3). The decrease in neutralizing antibody titer was statistically significant for all three variants (*p* < 0.0001) with respect to the wt strain.

## 4. Discussion

Recently, different variants of SARS-CoV-2 with mutations in the S protein have emerged, raising concerns about the protection elicited by natural infection or conferred by vaccination.

In this study, we evaluated the neutralizing activity against B.1.1.7, P.1, and B.1.351 VOCs in a panel of human serum samples from hospitalized patients infected by SARS-CoV-2 between March and May 2020 during the first wave of the pandemic in Italy [23].

Although we assessed the neutralizing activity at three time points for each patient, namely hospital admission (baseline), neutralizing antibody titer peak, and the last time point available (at discharge or decease), here we mainly discuss the last one since it may represent the “antibody baggage”.

In our study, 59.5% of the patients had a decrease (≥2-fold) in neutralizing antibody titer against B.1.1.7, and 21.4% showed a 4-fold decrease, as shown in a similar study [26]. On the contrary, other studies have reported that samples from convalescent or hospitalized patients are able to neutralize the B.1.1.7 variant, either maintaining similar levels of neutralizing activity or exhibiting a modest decrease compared to the original antibody titer [27,28,29]. We can hypothesize that the decrease in neutralizing antibody titer found in our study may be ascribed to the wt virus used, lacking D614G mutation. In general, the modest change in the potency of neutralization against this variant could be related to the N501Y mutation that, even though related to an increased binding to the ACE-2 receptor, does not appear to have a significant implication for neutralizing activity [28].

Regarding the P.1 variant, 83.3% of the patients involved in the study showed significantly reduced neutralizing activity (a ≥2-fold decrease), and 16.7% had a decrease higher than 4-fold, almost twice that reported for B.1.1.7. The reduction in neutralizing activity was even higher against the B.1.351 variant, with 90.5% of patients showing a fold decrease of ≥2 with a mean decrease almost 3 times higher than that against the P.1 variant. These findings are consistent with other studies [27,28,30,31] in which a reduced neutralizing potency of antibodies against these variants has been reported, although convalescent and vaccinated sera seem to better neutralize P.1 than B.1.351. The reduced neutralization could be ascribed to the E484K mutation shared between P.1 and B.1.351 and related to a possible escape from neutralizing antibodies. Considering that some mutations of P.1 are in common with B.1.351, such as the E484K mutation, it is very likely that a similar or higher reduction in neutralizing antibodies may be caused by variants carrying the same mutation. The immunoevasion could lead to reinfection of subjects who recovered from a previous infection [16] and could reduce the protection induced by vaccination and/or natural infection [27]. However, recent studies have proven that boosting pre-existing immunity against SARS-CoV-2 with one dose of mRNA vaccine induces a strong neutralizing response even against divergent variants (i.e., B.1.351) [8,32]. In Italy, subjects naturally infected by SARS-CoV-2 receive just one dose of vaccine from 3 to 6 months after infection [33]. In addition, a recent study [34] reported that neutralizing antibodies after natural infection by SARS-CoV-2 can last for at least 9 months.

Notably, one serum sample was not able to neutralize the B.1.1.7 variant, three were not able to neutralize the P.1 variant, and nine were not able to neutralize the B.1.351 variant, suggesting that symptomatic subjects previously infected by SARS-CoV-2 wt virus might not be fully protected against the emerging variants.

A key strength of the study is that the neutralizing activity has been assessed using authentic live SARS-CoV-2 viruses and not surrogate VN assay.

This study has some limitations. First, the cohort only included hospitalized patients, which may not be representative of the general population. The lack of follow-up samples after discharge does not allow the assessment of the full breadth of cross-neutralization potential that can be reached following more extended affinity maturation.

Overall, our findings provide evidence of a remarkable lower neutralization capacity of SARS-CoV-2 antibodies acquired during the first wave of the pandemic against the B.1.351 variant, and to a lesser extent against P.1 and B.1.1.7. This might suggest the possibility of reinfection or that previously infected individuals may be partially protected against current and new emerging variants with relevant mutations. However, the immune response induced by natural infection may protect from severe disease [35]. The emergence of the Delta VOC is also raising concern; however, so far, it appears that vaccines remain effective, especially after two doses [21,22,36]. Our study highlights the importance of evaluating SARS-CoV-2 pre-existing immunity against emerging variants as a tool to foresee the immune escape and the extent of vaccine efficacy.

## Figures and Tables

**Figure 1 viruses-13-01347-f001:**
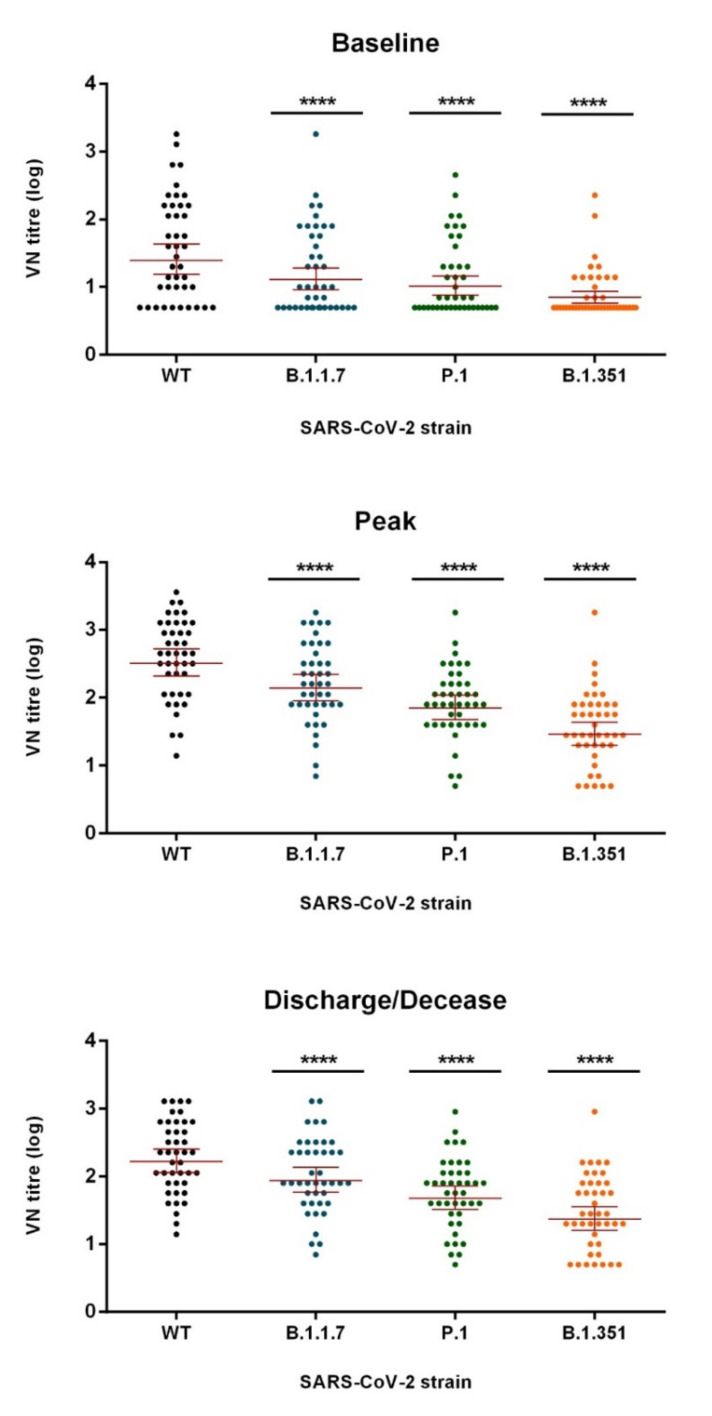
SARS-CoV-2 virus-neutralization (VN) titers with geometric mean and 95% confidence intervals, by time point and strain. For each variant, VN titers were compared with respect to the wt strain using a paired *t*-test (**** *p* ≤ 0.0001).

**Table 1 viruses-13-01347-t001:** Fold decrease in neutralizing antibody titers of B.1.1.7 (a), P.1 (b) and B.1.351 (c) variants with respect to the wt strain by time point (baseline, peak, and discharge/decease).

(a) B.1.1.7 Variant						
	Baseline		Peak		Discharge/Decease	
	N	%	N	%	N	%
-	19	45.2%	16	38.1%	17	40.5%
2-fold	12	28.6%	14	33.3%	16	38.1%
4-fold	7	16.7%	8	19.0%	9	21.4%
>4-fold	4	9.5%	4	9.5%	0	0.0%
Total	42	100%	42	100%	42	100%
Mean fold	3.0		3.5		2.3	
(SD)	(3.1)		(5.0)		(1.4)	
**(b) P.1 Variant**						
	**Baseline**		**Peak**		**Discharge/Decease**	
	**N**	**%**	**N**	**%**	**N**	**%**
-	12	28.6%	5	11.9%	7	16.7%
2-fold	15	35.7%	10	23.8%	14	33.3%
4-fold	6	14.3%	12	28.6%	14	33.3%
>4-fold	9	21.4%	15	35.7%	7	16.7%
Total	42	100%	42	100%	42	100%
Mean fold	4.3		6.0		4.4	
(SD)	(4.5)		(4.3)		(3.4)	
**(c) B.1.351 Variant**						
	**Baseline**		**Peak**		**Discharge/Decease**	
	**N**	**%**	**N**	**%**	**N**	**%**
-	10	23.8%	4	9.5%	4	9.5%
2-fold	10	23.8%	5	11.9%	9	21.4%
4-fold	2	4.8%	5	11.9%	6	14.3%
>4-fold	20	47.6%	28	66.7%	23	54.8%
Total	42	100%	42	100%	42	100%
Mean fold	10.2		18.5		11.0	
(SD)	(14.9)		(18.9)		(12.3)	

– means a decrease less than 2-fold; SD: standard deviation.

## Data Availability

Data is contained within the article.

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
