# Peer review of "Serum Neutralizing Activity against B.1.1.7, B.1.351, and P.1 SARS-CoV-2 Variants of Concern in Hospitalized COVID-19 Patients"

_viruses, 2021, doi:10.3390/v13071347_

Round 1

Reviewer 1 Report

Trombetta et al provide new data on the cross-neutralization of SARS-CoV-2 variants of concern B.1.1.7, B.1.351 and P1 (first identified in the UK, South Africa and Brazil respectively). Plasmas were obtained from a cohort of hospitalized patients from the first wave of covid-19 pandemic in Italy. The authors determined neutralization titers with a replicative virus isolate based assay measuring cytopathic effect by microscope. This data confirms previous reports showing a significant decrease in neutralization against B.1.351 and  P1 variants compared to Wuhan. They also observed a decrease in neutralization against B.1.1.7 which has not been uniformly reported when compared to Wuhan virus. While the results are not really novel and do not expand on previously published data, the experiments seem to have been performed correctly. Some improvements are necessary in the manuscript. Please find below a list of comments that follows the order of the manuscript.

Introduction:

  • The introduction is relatively well written, however the references are not always well selected. I think there is an excessive use of references to governmental reports when published peer reviewed work could be cited instead. Additionally, some citations do not match. For example ref 5 does not quite support the notion that B.1.1.7 is associated with an increase in fatality. For this, the author could cite another paper from the same group:

Davies, N.G.; Jarvis, C.I.; Edmunds, W.J.; Jewell, N.P.; Diaz-Ordaz, K.; Keogh, R.H. Increased mortality in community-tested cases of SARS-CoV-2 lineage B.1.1.7. Nature 2021, doi:10.1038/s41586-021-03426-1.

  • Considering the format of the paper (communication) the description of each variant could be shortened/synthetized.
  • There is an apparent misunderstanding of the authors regarding the D614G mutation. The authors mentioned this mutation specifically for the P1 variant, however, this mutation is present on all the variants. In fact D614G mutation appeared and expanded early during the first wave of the pandemic and became almost ubiquitous since. This should be corrected.

Methods:

  • There is no description of the WT/Wuhan virus used in the experiments. 2019-nCov/Italy-INMI1 is mentioned elsewhere in the text but this should be clearly stated in the Methods. This is also to make it clear that the virus used as WT virus is a D614 virus (and not a G614 one) as this can influence the interpretation of the data.
  • The authors should explain whether the different viruses (including Wuhan) were all tested at the same time within the same assay? This is recommended to get an accurate comparison of neutralization between the viruses and make sure that there is no deviation due to inter assay variability.
  • I did not understand why the methods contain a detailed “Sequencing methods”.
  • The statistics description is incomplete: the authors should at the very least confirm that they are using a paired T-Test.

Results:

  • It would seems more logical if Figure 1 appeared before table 1 as Figure 1 contains the raw data analyzed in table 1.
  • I think the “baseline titers” data do not bring much value to the study and could potentially be removed.
  • The legend of Figure 1 could be expanded a bit. Explain the statistical comparisons better. 

Discussion:

  • The discussion should not describe, again, some details of the results.
  • The authors could discuss further, the reduced neutralization of B.1.1.7 and how this observation differs from some other studies. Another study that also show a decrease is not cited (they also use a replicative virus): 

Supasa, P.; Zhou, D.; Dejnirattisai, W.; Liu, C.; Mentzer, A.J.; Ginn, H.M.; Zhao, Y.; Duyvesteyn, H.M.E.; Nutalai, R.; Tuekprakhon, A.; et al. Reduced neutralization of SARS-CoV-2 B.1.1.7 variant by convalescent and vaccine sera. Cell 2021, doi:10.1016/j.cell.2021.02.033.

  • The authors should discuss the limitations of the study, for example:
    • the cohort only contains hospitalized participants
    • the time of plasma collection which was performed, at the latest, at hospital discharge and therefore, does not necessarily indicate the full breath of cross-neutralization potential that can be reached by these patients following more extended affinity maturation.

General comments:

  • I would highly encourage the authors to name the different variants (in the text and in the figures) with their pango lineage names (B.1.17, B.1.351, P1) and not systematically refer to their country of initial identification (this can be mentioned in the intro once).
  • There is a need for further spell checking and some sentences should be simplified to help the reading (ex: sentence starting at line 263).

Author Response

Dear Editors and Reviewers,

Thank you for your comments and suggestions on how to improve the manuscript.

We have reviewed the manuscript according to your comments.

Comments and Suggestions for Authors

Trombetta et al provide new data on the cross-neutralization of SARS-CoV-2 variants of concern B.1.1.7, B.1.351 and P1 (first identified in the UK, South Africa and Brazil respectively). Plasmas were obtained from a cohort of hospitalized patients from the first wave of covid-19 pandemic in Italy. The authors determined neutralization titers with a replicative virus isolate based assay measuring cytopathic effect by microscope. This data confirms previous reports showing a significant decrease in neutralization against B.1.351 and  P1 variants compared to Wuhan. They also observed a decrease in neutralization against B.1.1.7 which has not been uniformly reported when compared to Wuhan virus. While the results are not really novel and do not expand on previously published data, the experiments seem to have been performed correctly. Some improvements are necessary in the manuscript. Please find below a list of comments that follows the order of the manuscript.

Response: We thank the reviewer for taking the time to review our manuscript and for the comments to improve the manuscript.

Introduction:

The introduction is relatively well written, however the references are not always well selected. I think there is an excessive use of references to governmental reports when published peer reviewed work could be cited instead. Additionally, some citations do not match. For example ref 5 does not quite support the notion that B.1.1.7 is associated with an increase in fatality. For this, the author could cite another paper from the same group:

Davies, N.G.; Jarvis, C.I.; Edmunds, W.J.; Jewell, N.P.; Diaz-Ordaz, K.; Keogh, R.H. Increased mortality in community-tested cases of SARS-CoV-2 lineage B.1.1.7. Nature 2021, doi:10.1038/s41586-021-03426-1.

Response: Thanks for your suggestion. When appropriate, published papers have been added (lines 47, 60). Reference 5 is related to the increased transmission of B.1.1.7 variant more than an increase in fatality. The suggested reference has been added to the increased risk death (line 50).

Considering the format of the paper (communication) the description of each variant could be shortened/synthetized.

Response: The description of the variants has been shortened (lines 55-59, 61, 69-70).

There is an apparent misunderstanding of the authors regarding the D614G mutation. The authors mentioned this mutation specifically for the P1 variant, however, this mutation is present on all the variants. In fact D614G mutation appeared and expanded early during the first wave of the pandemic and became almost ubiquitous since. This should be corrected.

Response: The reference to D614G mutation has been deleted (line 66).

Methods:

There is no description of the WT/Wuhan virus used in the experiments. 2019-nCov/Italy-INMI1 is mentioned elsewhere in the text but this should be clearly stated in the Methods. This is also to make it clear that the virus used as WT virus is a D614 virus (and not a G614 one) as this can influence the interpretation of the data.

Response: We apologize for the lack of information. A brief description of the WT/Wuhan virus has been added (lines 122-124). As specified in the Materials and Methods section, the used strain did not carry the S protein amino acid change D614G (line 124).

The authors should explain whether the different viruses (including Wuhan) were all tested at the same time within the same assay? This is recommended to get an accurate comparison of neutralization between the viruses and make sure that there is no deviation due to inter assay variability.

Response: Our SOPs and quality assurance do not allow operators to manage different pathogens under biological cabinet at the same time in order to prevent cross-contamination. However, in order to reduce inter-assay variability, all serum samples have been tested at the same time e by the same operator for each SARS-CoV-2 virus (e.g. all serum samples have been tested against Wuhan strain in the same run by the same operator). We hope this addresses your comment.  

I did not understand why the methods contain a detailed “Sequencing methods”.

Response: The sequencing methods section has been removed.

The statistics description is incomplete: the authors should at the very least confirm that they are using a paired T-Test.

Response: Thanks for the remark. We used a paired t-test to compare antibody levels for each variant with respect to the wild type virus. Indication on statistical analysis has been added (lines 176-177) and indication on statistical significance has been updated in the figure 1. We hope this will be acceptable.

Results:

It would seems more logical if Figure 1 appeared before table 1 as Figure 1 contains the raw data analyzed in table 1.

Response: Thanks for your suggestion. The order of figure 1 and table 1 has been reversed.

I think the “baseline titers” data do not bring much value to the study and could potentially be removed.

Response: We appreciate the reviewer’s comment. However, we prefer to keep the baseline titers for completeness.

The legend of Figure 1 could be expanded a bit. Explain the statistical comparisons better.

Response: The legend of Figure 1 has been expanded, explanation on statistical comparison has been added.

Discussion:

The discussion should not describe, again, some details of the results.

Response: The discussion has been revised accordingly.

The authors could discuss further, the reduced neutralization of B.1.1.7 and how this observation differs from some other studies. Another study that also show a decrease is not cited (they also use a replicative virus):

Supasa, P.; Zhou, D.; Dejnirattisai, W.; Liu, C.; Mentzer, A.J.; Ginn, H.M.; Zhao, Y.; Duyvesteyn, H.M.E.; Nutalai, R.; Tuekprakhon, A.; et al. Reduced neutralization of SARS-CoV-2 B.1.1.7 variant by convalescent and vaccine sera. Cell 2021, doi:10.1016/j.cell.2021.02.033.

Response: The discussion has been revised accordingly and the suggested study has been cited (lines 260-265).

The authors should discuss the limitations of the study, for example:

the cohort only contains hospitalized participants

the time of plasma collection which was performed, at the latest, at hospital discharge and therefore, does not necessarily indicate the full breath of cross-neutralization potential that can be reached by these patients following more extended affinity maturation.

Response: As suggested, limitations of the study have been expanded (lines 297-300).

General comments:

I would highly encourage the authors to name the different variants (in the text and in the figures) with their pango lineage names (B.1.17, B.1.351, P1) and not systematically refer to their country of initial identification (this can be mentioned in the intro once).

Response: Thanks for your suggestion. Names of the variants have been edited accordingly throughout the manuscript.

There is a need for further spell checking and some sentences should be simplified to help the reading (ex: sentence starting at line 263).

Response: The manuscript has been revised accordingly. We hope the current version is acceptable.

Reviewer 2 Report

The paper from Trombetta et al. is interesting as it brings interesting data concerning the neutralization of the SARS-CoV-2 variants by patient serum.

However, an effort should be done to improve the content (especially introduction and results section). Introduction should be updated and talk about the different variants of concern and of interest around the world. There are several sources on the internet that can help the authors do that. In the results section, the tables should be reformated and better explained. Moreover, it would be very intersting to get more data. I understand that getting serua may be difficult, but only 42 patients were studied, it would be interesting to evaluate the neutralization on more samples to get a better picture,a nd to evaluate other variants as well, including the Indian variant, which is also of concern.

Was the sequencing done to make sure that the viruses were not mutated after a passage in vero cells? The authors do not explain this part. If nohat was the purpose of sequencing?  

The neutralization assay looks quite standard but did the authors evaluated it with a higher concentration of virus? 100 TCID 50 appears quite low.

The authors should include not only the average results in their tables but also the standard deviation.

Like the intro, the discussion should also be updated and talk about other variants as well.

Author Response

Dear Editors and Reviewers,

Thank you for your comments and suggestions on how to improve the manuscript.

We have reviewed the manuscript according to your comments.

Comments and Suggestions for Authors

The paper from Trombetta et al. is interesting as it brings interesting data concerning the neutralization of the SARS-CoV-2 variants by patient serum.

Response: We thank the reviewer for taking the time to review our manuscript and for the comments to improve the manuscript.

However, an effort should be done to improve the content (especially introduction and results section). Introduction should be updated and talk about the different variants of concern and of interest around the world. There are several sources on the internet that can help the authors do that. In the results section, the tables should be reformated and better explained. Moreover, it would be very intersting to get more data. I understand that getting serua may be difficult, but only 42 patients were studied, it would be interesting to evaluate the neutralization on more samples to get a better picture,a nd to evaluate other variants as well, including the Indian variant, which is also of concern.

Response: Thanks for your suggestion. The introduction has been revised accordingly (lines 77-81). Tables have been improved. Unfortunately, we were not able to collect other serum samples and to recover other viruses other than those used in this study.

Was the sequencing done to make sure that the viruses were not mutated after a passage in vero cells?   The authors do not explain this part. If nohat was the purpose of sequencing?

Response: The sequencing methods section has been removed.

The neutralization assay looks quite standard but did the authors evaluated it with a higher concentration of virus? 100 TCID 50 appears quite low.

Response: We appreciate the reviewer’s comment. However, the virus concentration used in the present study, equal to 100TCID50, is the concentration recommended by the WHO for Micro-Neutralization assay (WHO, Manual for the laboratory diagnosis and virological surveillance of influenza) and is considered as the standard concentration in many different studies and clinical trials. Indeed, the viral load equal to 100TCID50, in accordance with the empirical formula obtained by applying the Poisson distribution, should be equal to approximately 70 plaque-forming units (pfu), which represents the measure of the infectious viral particles in a certain volume of medium used in each well of the microplate. Clearly, this is valid if the same cell system is used, and the virus is able to form plaques on the cell monolayer. The standard viral concentration generally used for plaque reduction neutralization assay is between 70 and 50 viral particles.

The authors should include not only the average results in their tables but also the standard deviation.

Response: Thanks for your suggestion. Standard deviation has been added to tables.

Like the intro, the discussion should also be updated and talk about other variants as well.

Response: As reported above, the introduction has been updated. Discussion section has been revised; we have added some sentences regarding the new variants, but we did not discuss in detail since few studies have been published so far on the new variants and our data did not cover this topic. We hope this will be acceptable.

Round 2

Reviewer 2 Report

Congratulations, the paper is ok for publication